# Are Edentulousness, Oral Health Problems and Poor Health-Related Quality of Life Associated with Malnutrition in Community-Dwelling Elderly (Aged 75 Years and Over)? A Cross-Sectional Study

**DOI:** 10.3390/nu10121965

**Published:** 2018-12-12

**Authors:** Mieke H. Bakker, Arjan Vissink, Sophie L.W. Spoorenberg, Harriët Jager-Wittenaar, Klaske Wynia, Anita Visser

**Affiliations:** 1Department of Oral and Maxillofacial Surgery, University of Groningen, University Medical Center Groningen, 9700 RB Groningen, The Netherlands; a.vissink@umcg.nl (A.V.); ha.jager@pl.hanze.nl (H.J.-W.); a.visser@umcg.nl (A.V.); 2Department of Health Sciences, Community and Occupational Medicine, University of Groningen, University Medical Center Groningen, 9700 RB Groningen, The Netherlands; s.dereus@hzd.nu (S.L.W.S.); k.wynia@umcg.nl (K.W.); 3Research Group Healthy Ageing, Allied Health Care and Nursing, Hanze University of Applied Sciences, 9714 CA Groningen, The Netherlands

**Keywords:** community-dwelling, older adults, oral health, edentulousness, complete denture, malnutrition, health-related quality of life (HRQoL)

## Abstract

As the population ages, the risk of becoming malnourished increases. Research has shown that poor oral health can be a risk factor for malnutrition in institutionalized elderly. However, it remains unclear whether oral health problems, edentulousness and health-related quality of life also pose a risk for malnutrition in community-dwelling older adults. In this cross-sectional observational study, 1325 community-living elderly (≥75 years) were asked to complete questionnaires regarding nutritional status, oral status (edentulous, remaining teeth, or implant-supported overdentures), oral health problems, health-related quality of life (HRQoL), frailty, activities of daily living (ADL) and complexity of care needs. Univariate and multivariate logistic regression analyses were performed with nutritional status as dependent variable. Of the respondents, 51% (*n* = 521) were edentulous, 38.8% (*n* = 397) had remaining teeth and 10.2% (*n* = 104) had an implant-supported overdenture. Elderly with complex care needs were malnourished most frequently, followed by frail and robust elderly (10%, 4.5% and 2.9%, respectively). Malnourished elderly reported more frequent problems with chewing and speech when compared with well-nourished elderly (univariate analysis). However, multivariate analysis did not show an association between malnutrition and oral health problems and edentulousness, although HRQoL was associated with malnutrition (odds ratio (OR) 0.972, confidence interval (CI) 0.951–0.955). Based on the results of this cross-sectional study, it can be concluded that poor HRQoL is significantly associated with malnutrition; however, edentulousness and oral health problems are not.

## 1. Introduction

Worldwide, life expectancy is increasing [1]. This also applies to the northern region of the Netherlands; 30% of the regional inhabitants will be >65 years by 2020 [2]. Staying vital and healthy during aging is challenging for many elderly, as many chronic diseases (e.g., diabetes, mental disease, coronary artery disease, organ problems, cancer) and health-related problems (e.g., malnutrition) commonly develop [3,4]. Usually, more than one chronic disease is present in elderly, a condition known as multimorbidity. Multimorbidity is frequently accompanied by polypharmacy, i.e., the use of multiple medicines. Recent studies have shown that the prevalence of multimorbidity and polypharmacy rapidly increases with age [5,6].

Multimorbidity, polypharmacy, advanced age and frailty are associated with an increasing risk of becoming malnourished [7,8,9,10,11]. Among community-dwelling elderly aged ≥75 years, the prevalence of malnutrition is 2.6% and increases rapidly when elderly become institutionalized or hospitalized (13.8% and 38.7%, respectively) [7,8,9]. Preventing malnutrition is crucial in this vulnerable group; malnutrition is associated with lower activities of daily living (ADL), lower quality of life (QoL), longer hospital stay and rehabilitation, higher risk of falls, higher infection rates, poor wound healing and higher mortality rates [12,13,14,15,16].

The causes of malnutrition are multifactorial [17]. Oral health problems such as tooth loss, toothache and chewing complaints are mentioned as contributing factors to malnutrition, especially in institutionalized elderly [18,19]. In this context it should be noted that the oral health of institutionalized elderly is generally poor, and that this poor oral health is usually present at the time of admission [20]. This indicates that poor oral health develops before elderly are admitted to a nursing home. A recent study showed that community-dwelling elderly with remaining teeth or implant-supported overdentures are less frail and have a better QoL than edentulous elderly [21]. This raises the following two questions. In a community-dwelling population, (1) does retaining one’s own teeth or having an implant-supported overdenture at older age also limit the risk of being malnourished? (2) are oral health problems (e.g., masticatory problems and dental pain) and a low health-related quality of life (HRQoL) associated with the risk of being malnourished? To address these questions, we assessed whether oral status (being edentulous, having own teeth or having an implant-supported overdenture), oral health problems and low HRQoL are associated with malnutrition in community-dwelling elderly aged ≥75 years.

## 2. Materials and Methods

### 2.1. Study Design and Participants

We performed a cross-sectional study among community-dwelling eligible elderly (*n* = 1325) participating in Embrace (‘SamenOud’ [literally translated into English as ‘ageing together’]). These elderly were patients of general practitioners (GPs) enrolled in Embrace. For details see the extensive description of the program Embrace published elsewhere [22,23,24,25]. The Medical Ethical Committee of the University Medical Center Groningen, Groningen, the Netherlands, assessed the study proposal and concluded that formal approval was not required (reference METc2011.108). The study was performed in accordance with the principles expressed in the Declaration of Helsinki.

### 2.2. Procedure and Assessments

Between June 2015 and November 2015, demographic characteristics such as age, sex, marital status, living situation, education level, income and health (underlying diseases, use of drugs) were collected at baseline, along with data from four validated health-related questionnaires:
Frailty was assessed by the Groningen Frailty Indicator (GFI) [26]. This instrument assesses physical and psychological frailty among elderly. The total score ranges from 0–15, with a higher score indicating a higher level of frailty. Someone with a score of ≥5 was regarded as frail [26]. The INTERMED questionnaire for the Elderly Self-Assessment (*IM-E-SA)* was used to assess the complexity of care needs [27]. It consists of 20 questions in four domains: biological, psychological and social needs and healthcare. The total score ranges from 0–60, with a higher score indicating more need for complex care. Someone with a score of ≥16 was regarded as in need of complex care.The level of dependency in activities of daily living was assessed using the Katz-15 index [28]. This index includes six physical ADL items, seven instrumental ADL activities and two additional ADL items. The total score ranges from 0–15, with a higher score indicating more dependency in performing daily activities.EuroQoL-5D (EQ-5D) was used to assess health-related quality of life (HRQoL) [29]. It consists of five questions: mobility, self-care, pain, usual activities and psychological status. The total score ranges from 0–1, with a higher score indicating a better perceived HRQoL. The second part of the EQ-5D is a Visual Analogue Scale (VAS). This EQ-VAS was used to mark current health status on a 20 cm vertical scale, with end points of 0 and 100. A higher score indicated a better HRQoL.

### 2.3. Risk Profiles

Based on scores of IM-E-SA and GFI, participants were classified into three groups: robust elderly, frail elderly and elderly with complex care needs. Robust elderly were defined as not having complex care needs and low levels of frailty (IM-E-SA < 16 and GFI < 5). Frail elderly were defined as having a higher level of frailty, but low level of complex care needs (IM-ESA < 16 and GFI ≥ 5). Elderly with complex care needs were defined as having substantial and ongoing healthcare needs, often resulting from chronic illness or disabilities (IM-E-SA ≥ 16).

### 2.4. Nutritional Status, Oral Status and Self-Reported Oral Health

All 1325 participants within Embrace and with a baseline assessment received an additional questionnaire consisting of 10 questions related to nutritional status and 13 questions on oral status, oral health, dental care and oral function. In case a questionnaire was incomplete, elderly were telephoned and interviewed so they could complete the questionnaire. If completing a questionnaire was not possible, the participant was excluded from this study.

The nutritional status questionnaire included self-reported body length, current body weight, body weight both 1 and 6 months ago and ability to eat. Nutritional status was defined as being malnourished (according to the guidelines of the Dutch Malnutrition Steering Group) or well-nourished [30].Malnutrition was assessed according to the guidelines of the Dutch Malnutrition Steering Group, which states that malnutrition among elderly aged ≥75 years is defined by a set of risk indicators of malnutrition: a BMI < 20 kg/m^2^ and/or unintentional weight loss of >10% in 6 months and/or unintentional weight loss of >5% in 1 month [30].The oral status and oral health questionnaire [21] included presence or absence of own teeth, an implant-supported overdenture or conventional denture. Oral health was assessed by presence or absence of problems related to pain or dry mouth, oral function (masticatory and speech problems) and oral self-care (cleaning habits, dental visits). In addition, participants were asked to rate their satisfaction with their oral status on a 10-point scale, ranging from 0 (very poor) to 10 (very good). Previous research showed that elderly experienced no problems with completing these questionnaires [21].

### 2.5. Statistics

SPSS IBM Statistics version 23.0 (SPSS, Chicago, IL, USA) was used for statistical analysis of the results. Chi-square tests and Fisher’s exact tests were used to analyze differences between subgroups risk profile and oral status. Demographic variables, oral status, risk profiles, general health and oral health were analyzed for differences between malnourished and well-nourished elderly using Mann-Whitney U-tests and Chi-square tests. For non-normally distributed variables median and interquartile ranges (IQR) were reported as measures of dispersion. A *p*-value < 0.05 was defined as statistically significant. If more than two groups were compared (e.g., oral status, risk profile), the Fisher-Freeman-Halton test was applied. Post hoc analysis per group was performed with Mann-Whitney *U*-tests or Chi-square tests, depending on normally or non-normally distributed variables.

Univariate logistic regression models were constructed to determine the odds ratio (OR) between the dependent variable (nutritional status) and independent variables, i.e., demographics (education and marital status), oral health (chewing problems, speech problems, eating problems) and general health (Katz-15 and EQ-5D).

A multivariate logistic regression was used to control for a confounding effect. In this model, the statistically significant independent variables (*p* < 0.05) of the univariate logistic regression model were entered in the multivariate analysis. Adjusted OR and corresponding 95% confidence intervals (CI) were determined. The Wald test (*p* < 0.05) was used to determine whether the effect was significant. Multicollinearity was tested and was regarded a problem when Tolerance was <0.1 or the variance inflation factor >10. These values were not seen for our variables, but after careful consideration it was decided to enter only the GFI and IM-E-SA scores, while the risk profiles (based on the scores of GFI and IM-E-SA) were not entered in the multivariate model to prevent the incorrect interpretation of multivariate analysis.

## 3. Results

### 3.1. Respondents

All 1325 elderly were eligible and were invited to participate (Figure 1) in this study. A total of 78.6% (*n* = 1041) gave their consent and returned the questionnaires. Out of 284 not participating patients, 18.8% (*n* = 249) were not willing to complete the questionnaires and 2.6% (*n* = 35) did not participate for unknown reasons. Another 1.4% (*n* = 19) had to be excluded due to missing or incomplete data. This resulted in a total of 1022 participating elderly (response rate 77.1%).

### 3.2. Patient Characteristics and Nutritional Status

Table 1 shows the patient characteristics and differences in characteristics between malnourished and well-nourished participants. In total, 4.8% of the participants were malnourished. Significantly more elderly in the malnourished group lived alone or were single and had a low education level compared to those in the well-nourished group.

Higher scores of GFI and IM-E-SA were found among malnourished elderly. Katz-15 scores were higher, while EQ-5D and EQ-VAS were significantly lower in the malnourished group. Complaints with chewing, eating hard foods and speech problems were reported significantly more often by malnourished elderly.

### 3.3. Risk Profiles and Malnutrition

Nutritional levels differed significantly between risk profiles (Table 1**).** To gain further insight into this observation, risk profiles were defined as dependent variables in Table 2. Based on their levels of frailty and need for complex care, participants were assigned to the robust, frail and complex care needs groups. The robust group consisted of 579 participants (56.7%), the frail group of 224 (21.9%) and the complex care needs group of 218 (21.4%). The robust group consisted of significantly more participants with remaining teeth and fewer edentulous elderly than the frail and complex group. Malnutrition was most frequent in the complex group (10%) when compared to the frail (4.5%) and robust (2.9%) groups.

### 3.4. Univariate and Multivariate Logistic Regression Analysis

Table 3 shows results of the univariate and multivariate logistic analysis with nutritional status as dependent variable. The univariate analysis showed that marital status and education were both associated with nutritional status. Single-living (OR 2.176) and a low education level (OR 2.116) showed a higher risk of malnutrition. No significant differences in oral status were found. The risk profiles robust and complex (OR 3.692) showed a statistically significant difference in nutritional status (*p* ≤ 0.001). Higher GFI, IM-E-SA and Katz-15 scores and lower EQ-5D and EQ-VAS scores showed an increased risk for malnutrition.

The multivariate analysis using nutritional status as a dependent variable is shown in the right column. When controlling for confounding variables in the model, only the EQ-VAS (HRQoL) remained statistically significant as a risk factor for malnutrition (OR 0.972, 95% confidence interval (CI) 0.951–0.995; *p* = 0.015).

## 4. Discussion

This study focused on malnutrition and associating factors among community-dwelling elderly aged ≥75 years. We found a general prevalence of malnutrition of about 5% for community-dwelling elderly, which is in accordance with previous research [8,31]. Oral health complaints were reported more frequently by malnourished elderly. However, in a multivariate model, oral health complaints and edentulism were not significantly associated with malnutrition, while a low HRQoL was.

The prevalence of malnutrition was higher in complex care elderly than in robust and frail elderly. This higher prevalence of malnutrition in elderly with complex care needs was associated with their greater number of comorbidities and substantial healthcare needs [32,33]. This might be due to the fact that robust elderly had a better general health (i.e., less polypharmacy, fewer comorbidities), a more independent ADL and a higher QoL when compared to elderly with complex care needs [21]. These more favorable conditions probably resulted in a more resilient health status for the robust elderly, which made them less vulnerable to potential health risks such as malnutrition. Frail elderly were less independent than robust elderly; however, they did not appear to be at a greater risk for malnutrition. Elderly with complex care needs already have to cope with deteriorating general health and a more dependent ADL level, and therefore being at higher risk for malnutrition.

Complaints about chewing, eating hard foods and speech problems were reported significantly more often by malnourished elderly. However, the multivariate analysis malnutrition did not show a significant association with oral health, which might seem to be inconsistent, as chewing problems and edentulousness and malnutrition are often related [34]. This lack of a significant association might be due to interactions amongst variables. This issue was also mentioned by El Osta et al. [35]. They reported that tooth loss and loss of functional tooth units (FTU) resulted in a higher risk for malnutrition among older adults. Similar to our study, their univariate analysis revealed that the subjective oral health indicators, prosthetic status and FTUs were statistically associated with malnutrition, while oral status was no longer an independent risk factor when applying a multivariate analysis. The number of FTUs could not be taken into account, as this study was based on self-reported data. The edentulous elderly were those elderly who reported an absence of all their teeth. The Dutch health insurance reimburses most of the costs for a complete denture. Therefore, it is standard care in the Netherlands that edentulous patients are provided with a complete denture. However, it is unclear how often dentures are worn. Sometimes, only the upper denture is worn or the denture is worn for a limited amount of time during the day [36]. We would suggest future research to focus on the number of FTUs (especially during eating), next to oral status and oral health.

HRQoL, determined by the EQ-VAS, showed a significant association with malnutrition in both the univariate and multivariate models. Previous research showed that 28% of the variability of HRQoL can be explained by the Oral Health Related Quality of Life (OHRQoL) [37]. Specific oral health-related problems, i.e., speech and chewing problems, were reported in this study and showed a significant association initially, and may have affected the OHRQoL and ultimately the HRQoL. An interesting topic for future research would be to determine the influence of these reported oral health problems on both OHRQoL and HRQoL, and their effect on malnutrition. Additional research within large groups of malnourished elderly regarding oral health and oral status would also be of interest.

### 4.1. Strengths and Limitations

The mains strengths of the study were the large study population and high response rate. The study population provided credible insight into general and oral health of elderly living at home. It is a good representation of the current population of community-dwelling elderly and both their oral and general health status.

Respondents differed significantly from non-respondents in regards to age, education, living status, income and polypharmacy (data not shown). The non-respondents were older, had a lower education level and lower monthly income, lived more often in a sheltered community and used more medication. Only demographic characteristics of the non-respondents were available. 

A limitation is related to the low prevalence of malnutrition (≈5%) in the elderly assessed, resulting in a low predictive value for the defined associated factors. The small numbers of elderly with malnutrition may have influenced extrapolation of the univariate and multivariate logistic analyses.

Finally, our study was a cross-sectional study, and since malnutrition can be a temporary state, the results should be interpreted with this in mind. Therefore, future research should focus on a larger group of community-dwelling elderly who are followed for a specific period (cohort study) in which the effect of oral status, oral health problems and (oral) HRQoL on nutritional status can be observed over time. Furthermore, following a population over time enables the determination of risk factors for malnutrition instead, and not just associating factors.

### 4.2. Clinical Implication

Malnutrition is usually related to a decline in general health in elderly. Although our study did not show that edentulism is associated with malnutrition, malnutrition is associated with poor HRQoL.

Maintaining good oral health (absence of pain, inflammation and tooth decay) and oral function (chewing ability and aesthetics) are presumably a relevant contributing factor to maintaining a high level of OHRQoL and HRQoL. Therefore, care professionals should focus on maintaining good oral health and a high HRQoL.

## 5. Conclusions

Based on the results of this cross-sectional study, edentulousness and self-reported oral health problems are not associated with malnutrition; however, a poor HRQoL is.

## Figures and Tables

**Figure 1 nutrients-10-01965-f001:**
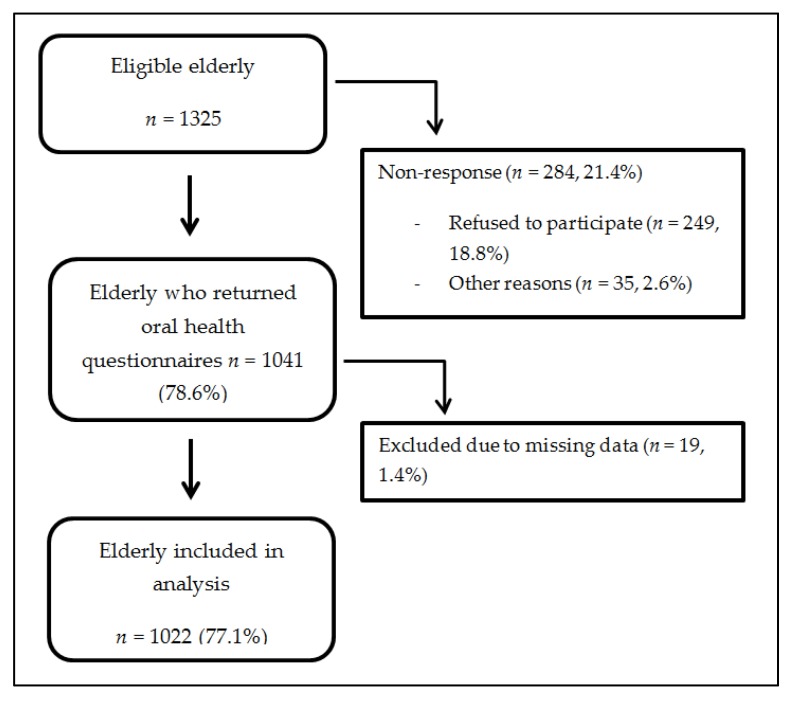
Flow diagram of patient inclusion process.

**Table 1 nutrients-10-01965-t001:** Nutritional status and patient-characteristics.

	Total (*n* = 1022)	Malnourished (*n* = 49, 4.8%)	Well-Nourished (*n* = 973, 95.2%)	*p*-Value between Nutritional Status
Demographics				
Age (median, interquartile range (IQR))	80 (77–84)	80 (77–84)	81 (79–85)	0.208
Sex (women, *n* (%))	598 (58.6)	35 (71.4)	563 (57.9)	0.074
Marital status-widow, divorced, single (*n* (%))	460 (45.1)	31 (63.3)	429 (44.2)	0.009
Sheltered home (*n* (%)) ^a^	103 (10.1)	7 (14.3)	96 (9.9)	0.327
Low education level (*n* (%)) ^b^	415 (40.7)	28 (58.2)	387 (39.8)	0.011
Low income (*n* (%)) ^c^	319 (39.3)	18 (48.6)	301 (38.9)	0.235
Oral status				
Remaining teeth (*n* (%))	397 (38.8)	20 (40.8)	377 (38.7)	0.772
Edentulous (*n* (%))	521 (51)	24 (49)	497 (51.1)	0.774
Implant-supported overdenture (*n* (%))	104 (10.2)	5 (10.2)	99 (10.2)	1.00
Total oral status	1022 (100)	49 (100)	973 (100)	0.932
Risk profile				
Complex (*n* (%))	219 (21.4)	22 (44.9)	197 (20.2)	0.000
Frail (*n* (%))	224 (21.9)	10 (20.8)	214 (22.0)	0.850
Robust (*n* (%))	579 (56.7)	17 (35.4)	562 (57.8)	0.002
Total risk profile	1022 (100)	49 (100)	973 (100)	≤0.001
General health				
Polypharmacy (*n* (%)) ^e^	582 (56.9)	32 (66.7)	550 (56.6)	0.168
Number of chronic conditions (median, IQR) ^d^	2 (1–3)	2 (1–4)	2 (1–3)	0.761
Frailty (GFI, median, IQR) ^f^	4 (2–6)	5 (3–7)	4 (2–6)	≤0.001
Complex care (IM-E-SA, median, IQR) ^g^	10 (6–15)	13 (9–18.5)	10 (6–14)	0.001
Activities of daily living (Katz-15, median, IQR) ^h^	1 (0–3)	2 (0–4)	1 (0–3)	0.004
Health-related quality of life (EQ-5D, median, IQR) ^i^	0.807 (0.693–0.861)	0.775 (0.610–0.843)	0.807 (0.719–0.893)	0.004
Health-related quality of life (EQ-VAS, median, IQR) ^j^	75 (60–80)	60 (50–72.5)	75 (60–80)	≤0.001
Oral health				
Irregular dental visits (*n* (%)) ^k^	483 (47.3)	21 (42.9)	462 (47.5)	0.527
Poor oral hygiene (*n* (%)) ^l^	33 (3.2)	4 (8.2)	33 (3.2)	0.068
Chewing problems (*n* (%)) ^m^	116 (11.4)	12 (24.5)	104 (10.7)	0.003
Eating problems (*n* (%)) ^m^	27 (2.6)	5 (10.2)	22 (2.3)	0.008
Speech problems (*n* (%)) ^m^	7 (0.7)	2 (4.1)	5 (0.5)	0.041
Recent history of dental pain (<6 months) (*n* (%))	106 (10.4)	8 (16.3)	98 (10.1)	0.154
Dry mouth during day or night (*n* (%)) ^m^	222 (21.7)	15 (30.6)	207 (21.5)	0.124
Dry mouth during day (n (%)) ^m^	78 (7.6)	3 (6.1)	75 (7.7)	1.000
Dry mouth during night (n (%)) ^m^	202 (19.8)	13 (26.5)	189 (19.5)	0.226
Insecurity (n (%)) ^m,n^	18 (1.8)	2 (4.1)	16 (1.6)	0.213
Satisfaction (median, IQR) ^o^	8 (7–8)	8 (7–8)	8 (7–8)	0.410

^a^ Sheltered home: living in a sheltered accommodation or a home for the elderly. ^b^ Low education level: (less than) primary school or low vocational training. ^c^ Low income: <€1450 per month. ^d^ Number of chronic diseases: total number of present chronic diseases out of listing of 18 chronic diseases (e.g., diabetes mellitus, osteoporosis). ^e^ Polypharmacy: the use of more than four drugs. ^f^ GFI: Groningen Frailty Indicator. ^g^ IM-E-SA: INTERMED for the Elderly Self-assessment. ^h^ Katz-15: Katz extended. ^i^ EQ-5D: EuroQoL-5D. ^j^ EQ-VAS: EuroQoL Visual Analogue Scale. ^k^ Irregular dental visit: not visiting a dentist over the past 2 years. ^l^ Poor oral hygiene: not brushing at least once a day. **^m^** Complaint is ‘often’ or ‘very often’. ^n^ Feeling insecure or ashamed about oral status. ^o^ Feeling satisfied with oral status (range 0–10, higher score means more satisfied).

**Table 2 nutrients-10-01965-t002:** Overview of risk profiles and oral status and malnutrition.

	Total (*n* = 1022)	Complex (*n* = 219, 21.4%)	Frail (*n* = 224, 21.9%)	Robust (*n* = 579, 56.7%)	*p*-Value between Subgroups Risk Profile (Complex, Frail, Robust)
Oral status					
Remaining teeth (*n* (%))	387 (38.8)	76 (34.9) ^b^	71 (31.7) ^c^	249 (43)	0.005
Edentulous (*n* (%))	521 (51)	125 (57.3) ^b^	133 (59.4) ^c^	263 (45.4)	≤0.001
Implant-supported overdenture (*n* (%))	104 (10.2)	17 (7.8)	20 (8.9)	67 (11.6)	0.228
Total oral status (*n*, (%))	1022 (100)	219 (100)	224 (100)	579 (100)	0.002
Malnutrition					
Malnutrition (*n* (%))	49 (4.8)	22 (10) ^a,b^	10 (4.5)	17 (2.9)	≤0.001

^a^*p* < 0.05 Complex and frail elderly. ^b^
*p* < 0.05 Complex and robust elderly. ^c^
*p* < 0.05 Frail and robust elderly.

**Table 3 nutrients-10-01965-t003:** Univariate and multivariate logistic regression analysis using nutritional status as dependent variable.

	Univariate Logistic Regression Analysis				Multivariate Logistic Regression Analysis ^a^			
	B (SE)	OR ^b^	95% CI ^c^	*p*-Value	B (SE)	OR ^b^	95% CI ^c^	*p*-Value
Demographics								
Age	0.029	1.026	0.969–1.085	0.380				
Sex	0.272	1.816	0.965–3.419	0.065				
Marital status: single, widow, divorced	0.240	2.176	1.201–3.943	0.010	0.329	1.714	0.899–3.268	0.101
Living status: sheltered	0.422	0.658	0.287–1.504	0.321				
Low education level	0.300	2.116	1.175–3.810	0.012	0.325	1.691	0.895–3.195	0.105
Low income	0.337	1.489	0.769–2.882	0.238				
Oral status				0.955				
Edentulous (reference)		1	-	-				
Remaining teeth	0.310	1.099	0.598–2.018	0.762				
Implant-supported overdenture	0.504	1.046	0.390–2.807	0.929				
Risk profile				≤0.001				
Robust (reference)		1	-	-				
Complex	0.333	3.692	1.921–7.096	≤0.001				
Frail	0.407	1.545	0.696–3.427	0.285				
General health								
Chronic conditions ^d^	0.077	1.072	0.922–1.246	0.366				
Polypharmacy ^e^	0.313	0.652	0.353–1.203	0.171				
Frailty (GFI) ^f^	0.050	1.218	1.106–1.342	≤0.001	0.085	1.068	0.904–1.262	0.437
Complex care (IM-E-SA) ^g^	0.021	1.070	1.027–1.115	0.001	0.038	0.973	0.903–1.049	0.481
ADL (Katz-15) ^h^	0.047	1.157	1.055–1.267	0.002	0.078	0.977	0.839–1.137	0.763
Health-related quality of life (EQ-5D) ^i^	0.736	0.067	0.016–0.285	≤0.001	1.348	0.339	0.024–4.763	0.423
Health-related quality of life (EQ-VAS) ^j^	0.008	0.963	0.947–0.979	≤0.001	0.011	0.972	0.951–0.995	0.015
Oral health								
Irregular dental visits	0.296	0.830	0.465–1.481	0.527				
Poor oral hygiene	0.555	2.893	0.975–8.583	0.055				
Chewing problems	0.348	2.707	1.368–5.354	0.004	0.438	2.014	0.853–4.753	0.110
Eating problems	0.519	4.907	1.775–13.567	0.002	0.714	1.478	0.365–5.994	0.584
Speech problems	0.850	8.230	1.556–45.533	0.013	1.001	5.630	0.791–40.070	0.084
Recent dental pain	0.401	1.738	0.792–3.814	0.168				
Dry mouth during day or night	0.320	1.628	0.870–3.047	0.127				
Insecurity ^k^	0.765	2.540	0.567–11.370	0.223				
Satisfaction ^l^	0.115	0.888	0.709–1.112	0.301				

^a^*R*^2^ = 0.114 (Nagelkerke), 0.036 (Cox&Snell) *χ*^2^ 0.212. ^b^ OR: odds ratio. ^c^ 95% CI: 95% confidence interval. ^d^ Number of chronic diseases: total number of present chronic diseases out of listing of 18 chronic diseases (e.g., diabetes mellitus, osteoporosis). ^e^ Polypharmacy: use of more than four drugs. ^f^ GFI: Groningen Frailty Indicator. ^g^ IM-E-SA: INTERMED for the Elderly Self-assessment. ^h^ Katz-15: Katz extended. ^i^ EQ-5D: EuroQoL-5D. ^j^ EQ-VAS: EuroQoL Visual Analogue Scale. ^k^ Feeling insecure or ashamed about oral status. ^l^ Feeling satisfied with oral status (range 0–10, higher score is more satisfied).

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
