# Peer review of "Are Edentulousness, Oral Health Problems and Poor Health-Related Quality of Life Associated with Malnutrition in Community-Dwelling Elderly (Aged 75 Years and Over)? A Cross-Sectional Study"

_nutrients, 2018, doi:10.3390/nu10121965_

Reviewer 1 Report

Thi is an interesting manuscript based on a well-designed study. This reviewer encourages the authors to consider the following comments/suggestions in view of further improving the manuscript.

Abstract

Expand HRQoL first and then use the abbreviation

Introduction

Pg2 Ln52:

How about polypharmacy?

Also, the introduction would have been improved if authors could move a bit from local context to worldwide situation and cite recently published international studies which showed that multimorbidity as well as polypharmacy and nutritional risk are common among elderly in this age group. 

Suggested references:

1. Amarasena et al. Health behaviours and quality of life in independently living South Australians aged 75 years or older. Aust Dent J 2018;63(2):156-162. 

2. Keuskamp et al. General health, wellbeing and oral health of patients older than 75 years attending health assessments. Aust J Prim Health 2018;24:177-182.

Pg2 L60: ‘…..problems such as tooth loss, …’

Discussion:

Although the response rate was high there were about 22% non-respondents. These non-respondents may have been different from respondents in regards to basic characteristics and accordingly it might have affected the findings.  Did the authors carry out such a comparison? The authors are encouraged to mention this in their discussion perhaps under the strengths and limitations section. 

Author Response

Reviewer 1

* Remark of the reviewer: This is an interesting manuscript based on a well-designed study.

Our response: We thank the reviewer for his compliment.

* Remark of the reviewer: Abstract: Expand HRQoL first and then use the abbreviation

Our response: We have now written the abbreviation HRQoL in full first (Pg.1 Ln. 30).

* Concern of the reviewer: Introduction: Pg. 2 Ln. 52: How about polypharmacy?

Our response: We added the term polypharmacy next to multimorbidity as well as that we included two articles concerning polypharmacy among older adults.

 Revised text: Pg.2 Ln. 51  The following text was added to the introduction:

Multimorbidity is frequently accompanied by polypharmacy, i.e., the use of multiple medicines. Recent studies have shown that the prevalences of multimorbidity and polypharmacy rapidly increase with age. [5,6]

* Concern of the reviewer: Also, the introduction would have been improved if authors could move a bit from local context to worldwide situation and cite recently published international studies which showed that multimorbidity as well as polypharmacy and nutritional risk are common among elderly in this age group.

Suggested references:

1. Amarasena et al. Health behaviours and quality of life in independently living South Australians aged 75 years or older. Aust Dent J 2018;63(2):156-162.

2. Keuskamp et al. General health, wellbeing and oral health of patients older than 75 years attending health assessments. Aust J Prim Health 2018;24:177-182.

Our response: We added the suggested international studies to the introduction.

Revised text: Pg. 2 Ln. 54  We added to the text:

Multimorbidity, polypharmacy, advanced age, and frailty are associated with an increasing risk of becoming malnourished.[7-11]

 * Remark of the reviewer: Pg.2 Ln. 60: ‘…..problems such as tooth loss, …’

Our response: ‘Such’ was replaced by ‘such as’ as suggested. Pg.2 Ln. 60.

 * Concern of the reviewer: Although the response rate was high there were about 22% non-respondents. These non-respondents may have been different from respondents in regards to basic characteristics and accordingly it might have affected the findings. Did the authors carry out such a comparison? The authors are encouraged to mention this in their discussion perhaps under the strengths and limitations section.

Our response: We have added information regarding the non-respondents to the section Strengths and limitations. Only demographic characteristics of the non-respondents were available, which showed an older and less educated population with more medication among the non-respondents.

Revised text:

Pg. 10 Ln. 266. We have added to the Strengths and limitations:

Respondents differed significantly from non-respondents regarding age, education, living status, income and polypharmacy (data not shown). The non-respondents were older, had a lower education level and lower monthly income, lived more of in a sheltered community and used more medication. Only demographic characteristics of the non-respondents were available.

 Reviewer 2 Report

This is an interesting cross-sectional study. 

Table 2 is unclear, misalignment makes difficult to understand the distribution of oral health status by category of frailty/risk - what is the total referring to ? What interest ? Please specify that is not the number of implants per patients, rather the number (%) of patients with dental implants-supported prostheses. Please specify the statistical test used for the 3 groups comparisons and then for the 2x2. 

Table 2 and 3 are displaying the same results in two different ways. I suggest to rethink data presentation and merge into one table. Since it is a cross-sectional, both directions are valid. Moreover, being consistent with the exposure and outcome (independent and dependent variables) will help the reader. 

Table 4 and 5 should be merged as well. It will be interesting to see OR (95%CI) at the univariate and then multivariate on the same line. Please specify the co-variables included in the multivariate model. In alternative, univariate OR can be presented in the Table 1 (since the dependent variable is Malnutrition status). Due to the small sample size in the group of malnourished, the number of co-variates is limited. Please specify. 

Are edentulous individuals wearing a prosthesis ? How many are completely edentulous and not wearing any prosthesis? Indeed, it appears to have a greater impact the number of functional units (determined by natural teeth or prosthesis) than the number of teeth lost. Please detail more this important aspect in the discussion. 

Conclusion are based on univariate analysis in which the influence of many potential confounders and bias is present. It would be better to base the key message of the article on the results of the multivariate analysis. 

Author Response

Reviewer 2

* Remark of the reviewer: This is an interesting cross-sectional study.

Our response: We thank the reviewer for this kind comment.

* Concern of the reviewer: Table 2 is unclear, misalignment makes difficult to understand the distribution of oral health status by category of frailty/risk - what is the total referring to? What interest?

Please specify that is not the number of implants per patients, rather the number (%) of patients with dental implants-supported prostheses. Please specify the statistical test used for the 3 groups comparisons and then for the 2x2. 

Our response: We have added extra lines within Table 1 and Table 2 to provide a clear view which categories were combined to form the variables Oral status and Risk profile. Oral status and risk profiles are categorized into 3 categories. When the 3 groups of oral status and risk profiles are added vertically (columns), the total group (malnourished or well-nourished) is formed. We have added the term total oral status and total risk profile to emphasize that Oral status and Risk profiles consist of 3 categories. See also Pg. 5 Ln. 175.

We have replaced ‘Implants’ for ‘Implant-supported overdenture’ in all tables to specify it concerns the number of patients with overdenture instead of number of implants.

The statistical tests used for 3 group and 2x2 comparisons are mentioned in the Method section Statistics. We have now specified that the post hoc analysis was carried out per group. Pg. 4 Ln. 141:

If more than two groups were compared (e.g., oral status, case complexity), the Fisher-Freeman-Halton test was applied. Post hoc analysis per group was performed with Mann-Whitney U-tests or Chi-square tests, depending on normally or non-normally distributed variables.

 * Concern of the reviewer: Table 2 and 3 are displaying the same results in two different ways. I suggest to rethink data presentation and merge into one table. Since it is a cross-sectional, both directions are valid. Moreover, being consistent with the exposure and outcome (independent and dependent variables) will help the reader. 

Our response: We have tried to combine tables 2 and 3 into 1 table. Combining was only possible when we use malnutrition as dependent variable. However, such a change would result in the same outcomes as presented in table 1.

After careful consideration we have decided to remove table 3, as it does not provide any additional information, since oral status was not statistically significant in table 1.

We decided to preserve table 2 as it provides detailed information on the prevalence of malnutrition and risk profiles. Table 1 does not provide this information, but shows significant differences among the risk profiles.

Revised text: Pg. 6 Ln. 186.  We have added to the Results 3.2:

Nutritional status differed significantly between risk profiles (Table 1). To gain further insight into this observation, risk profiles were defined as dependent variables in Table 2

* Remark of the reviewer: Table 4 and 5 should be merged as well. It will be interesting to see OR (95%CI) at the univariate and then multivariate on the same line. Please specify the co-variables included in the multivariate model. In alternative, univariate OR can be presented in the Table 1 (since the dependent variable is Malnutrition status). Due to the small sample size in the group of malnourished, the number of co-variates is limited. Please specify. 

Our response: We have merged table 4 and 5 into Table 3 (Pg. 7 Ln. 210). In the new Table 3, we have included only those variables from the univariate regression analysis that were statistically significant (p<0.05). Next, the risk profiles were fully based on the score of GFI and IM-E-SA. Combining these factors in the multivariate model might cause interaction among the variables. Therefore, we decided to only add GFI and IM-E-SA score and not the risk profiles.

This approach was addressed in the Methods section Statistics Pg.4 Ln. 148 (A multivariate logistic regression was used to control for a confounding effect. In this model, the statistically significant independent variables (p<0.05) of the univariate logistic regression model were entered in the multivariate analysis. Adjusted OR and corresponding 95% confidence intervals were determined. The Wald test (p<0.05) was used to determine whether the effect was significant).

* Concern of the reviewer: Are edentulous individuals wearing a prosthesis? How many are completely edentulous and not wearing any prosthesis? Indeed, it appears to have a greater impact the number of functional units (determined by natural teeth or prosthesis) than the number of teeth lost. Please detail more this important aspect in the discussion. 

Our response: We thank the reviewer for this valuable comment. We have now added this important aspect in the Discussion. In this cross-sectional study based on self-reported data, it was not possible to determine the amount of FTUs per person, as it is possible that dentures are not worn, or only a limited amount of time.

Revised text: Pg. 9 Ln. 244

In this study the number of FTUs could not be taken into account as this study was based on self-reported data. The edentulous elderly were those elderly who reported the absence of all their teeth. The Dutch health insurance reimburses most of the costs for a complete denture. Therefore, it is standard care in the Netherlands that edentulous patients are provided with a complete denture. It is unclear, however, how often dentures are worn. Sometimes only the upper denture is worn or the denture is worn for a limited amount of time during the day.[36] We would suggest future research to focus on the number of FTU’s (especially during eating), next to oral status and oral health.

 * Concern of the reviewerConclusion are based on univariate analysis in which the influence of many potential confounders and bias is present. It would be better to base the key message of the article on the results of the multivariate analysis. 

Our response: We have removed the conclusions of the univariate analysis (‘problems with chewing and speech were more often reported by malnourished elderly than by well-nourished elderly´) in the following sections:

Abstract – Pg. 1 Ln. 39

Clinical implication – Pg. 10 Ln. 281     

Conclusions – Pg. 10 Ln. 290

 Round  2

Reviewer 1 Report

Authors have revised the manuscript as per the suggestions/comments made by the reviewer/s and consequently the manuscript has been considerably improved. The revised manuscript  can be accepted in the current form after the authors have attended to minor suggestions below:

Pg9, Ln248: "Sometimes occurs only...." Delete 'occurs'

Pg9, Ln266: Replace 'regarding' with 'in regards to'

Author Response

* Remark of the reviewer: Pg9, Ln248: "Sometimes occurs only...." Delete 'occurs'

Our response: The verb ‘occurs’ is deleted from this sentence. Pg.9, Ln.248.

 * Remark of the reviewer: Pg9, Ln266: Replace 'regarding' with 'in regards to'

Our response: ‘Regarding’ is replaced by ‘in regards to’ as suggested. Pg.9, Ln.266